# Sparking the Interest of Girls in Computer Science via Chemical Experimentation and Robotics: The Qui-Bot H_2_O Case Study [note 1]

**DOI:** 10.3390/s22103719

**Published:** 2022-05-13

**Authors:** Marta I. Tarrés-Puertas, Jose Merino, Jordi Vives-Pons, Josep M. Rossell, Montserrat Pedreira Álvarez, Gabriel Lemkow-Tovias, Antonio D. Dorado

**Affiliations:** 1Department of Mining, Industrial and ICT Engineering, Universitat Politècnica de Catalunya—BarcelonaTech (UPC), 08242 Manresa, Spain; jose.merino.millo@upc.edu (J.M.); jordi.vives-pons@upc.edu (J.V.-P.); toni.dorado@upc.edu (A.D.D.); 2Department of Mathematics, Universitat Politècnica de Catalunya—BarcelonaTech (UPC), 08242 Manresa, Spain; josep.maria.rossell@upc.edu; 3Research Group in Education, Neuroscience, Experimentation and Learning (GRENEA), Faculty of Social Sciences, UVic-UCC, 08241 Manresa, Spain; mpedreira@umanresa.cat (M.P.Á.); glemkow@umanresa.cat (G.L.-T.)

**Keywords:** computational thinking, STEM, robotics, diversity, inclusion, chemistry, gender, stereotypes, diversity, education

## Abstract

We report a new learning approach in science and technology through the Qui-Bot H_2_O project: a multidisciplinary and interdisciplinary project developed with the main objective of inclusively increasing interest in computer science engineering among children and young people, breaking stereotypes and invisible social and gender barriers. The project highlights the social aspect of robotics applied to chemistry, at early ages. We successfully tested the project activities on girls between 3 to 13 years old. After taking part in the project, the users rated their interest in science and technology to be higher than before. Data collected during experiences included background information on students, measurements of the project’s impact and students’ interest in it, and an evaluation of student satisfaction of this STEM activity. The Qui-Bot H_2_O project is supported by the actions of territorial public administrations towards gender equality and the contributions of humanistic and technological universities and entities which specialize in education and business.

## 1. Introduction

Qui-Bot H_2_O is an interdisciplinary research project about teaching innovation and the dissemination of science that brings together two subjects with high potential and high demand: chemistry and programming. By combining informatics and chemistry, Qui-Bot H_2_O is a multidisciplinary approach to robotics that aims to attract girls to and reduce the gender gap for informatics engineering. Scientists and engineers from disciplines such as chemistry, computer science, mathematics, and research in education work together on the Qui-Bot H_2_O project.

### 1.1. Goals

The main goal of this work is to develop an educational project which contributes to the incorporation and improvement of computational thinking in pre-school- as well as primary and secondary school-aged students through experimentation in chemistry and robotics via games and activities. Our work focuses on the design of the methodology and activities, testing, and validation to implement computational thinking in the different educational stages. The project includes the definition of a set of activities that seek to strengthen STEM (science, technology, engineering, and mathematics) competencies. The design of Qui-Bot activities takes advantage of the innate curiosity of boys and girls for scientific experiments from an early age and their eagerness to play with robots. In this way, the curricular contents in these subjects are complemented in the different educational stages by the involvement and complicity of the teaching staff. For this reason, the proposal is completed with training activities addressed to teachers and instructors to reinforce and guarantee success in the implementation of the project on a regular basis. Our approach involves designing and constructing a Lego-based pipetting robot with sensors, motors, and actuators that can reliably handle liquids and support a variety of science and chemistry experiments. We provide the software to monitor and program the robot, which allows students to learn computational thinking in an intuitive way. The robot is programmed via the new software interface, and the instructions to carry out scientific experiments are provided in real time. 

The actions of the Qui-Bot H_2_O project are aligned with the Sustainable Development Goals (SDG) that follow. SDG 4 focuses on Quality Education, and Qui-Bot activities support computational thinking via robotics and experimentation in the classroom in a playful manner, allowing students to solve real-life problems in a visual and intuitive way. Our approach is to provide a concrete tool (made up of hardware, software, and related activities) to be used directly in classrooms while enabling teachers to explain scientific concepts in an innovative way and at the same time encourage users to develop computational thinking skills. Teachers can adapt the tool to meet their specific needs. Furthermore, as it is a tool intended to be used in educational centers, it ensures inclusive and equitable quality education. Our work addresses SDG 5, Gender Equality, by empowering women and girls into technology from an early age so as to reduce invisible social barriers and stereotypes that limit what women think they are capable of. In relation to SDG 9, Industry, Innovation, and Infrastructure, the Qui-Bot project fosters inclusion and sustainable industrialization through the contextualization of activities to the real world and through the use of robotics and technology as key examples in industrialization. Regarding SDG 10, Reduced Inequalities, through the inclusion of computational thinking in robotics across the curriculum, children from all socioeconomic backgrounds can benefit from robotics education. At present, robotics is typically an extracurricular course and, unfortunately, is tied to social gender stereotypes. Our project’s objectives align well with the International Year of Creative Economy for Sustainable Development, 2021, as we promote collaboration between partners, provide opportunities for future lecturers and researchers, and foster innovation in education for all children, with a focus on girls.

### 1.2. Motivation

The great demand for professionals in ICT (information and communications technology) and industry in the coming years, according to national studies (see, for example, the InfoJobs report [1] and the Barcelona Chamber of Commerce, 2020 [2]), underlines the need for strategies to increase the number of professionals in STEM and to promote women in scientific careers. The gender gap in universities oriented towards STEM studies is persistent year after year. As stated in the latest report from the Spanish University system (Ministry of Universities, 2021 [3]), in the 2018–2019 academic year, women represented only 24.8% of people enrolled in engineering and architecture, 4% less than the previous year. 

A European Commission report in 2016 [4] predicted that employment in STEM occupations in the European Union (EU) (for all levels) will increase by 12.1% in 2025: a much higher rate than the projected increase of 3.8% for other occupations. Another European Commission study in 2018 [5] described how 57% of tertiary graduates in the EU are women, but only 24.9% of them graduate in ICT-related fields and very few enter the sector. Women make up 13% of graduates in ICT-related fields working in digital jobs, compared to 15% in 2011. All in all, according to experts, seven million jobs in Europe will have to be filled by science and technology by 2025.

According to the innovative UNESCO report in 2017 [6], only 35% of STEM students in higher education worldwide are women and differences are observed within STEM disciplines. For example, only 3% of higher education students choose information and communications technology (ICT) studies. This gender disparity is alarming, especially since STEM careers are often known as the jobs of the future, which drive innovation, social welfare, inclusive growth, and sustainable development. As stated in a UNESCO report in 2021 [7], it is necessary to stimulate interest from an early age, to combat stereotypes, to train teachers, to encourage girls to pursue STEM careers, to develop gender-sensitive curricula, to guide young girls and women, and to change their mindset. In agreement with a Science report in 2017 [8], stereotypes are endorsed by 6-year-olds and influence children’s interests. Specifically, 6-year-old girls are less likely than boys to believe that members of their gender are “really, really smart”. Also, at age 6, girls begin to avoid activities that are said to be for children who are “really, really smart”. These findings suggest that gender-based notions of “being bright”, as well as gender roles associated with traditional notions of male and female activities, are acquired early and have an immediate effect on children’s interests.

Our work is the result of a collaboration between universities specializing in technology, science, and pedagogy and economic and social agents. The group of collaborating entities has a common objective: the promotion of education and scientific technical vocations to children at an early age, especially those of the female gender, with the aim of reducing the imbalance between the need for professionals and the lack of vocations. 

Nowadays, there are numerous educational projects involving robotics, such as Lego Mindstorms or Lego competitions (see, for example, the works of [9,10,11]). These activities are focused on mechanical engineering projects and computer programming. However, there is currently a great national gap concerning the application of robotics to other disciplines such as chemistry. In this project, we seek to add significant value by combining the learning of chemistry and programming using robotics with the aim of attracting boys and girls to STEM studies at an early age. Our proposal arises from a very satisfactory international experience at Stanford University run by the Department of Bioengineering led by the researcher Ingmar H. Riedel-Kruse [12].

The project is supported by the accumulated experience in educational innovation initiatives in chemistry carried out by the Chemical Engineering faculty of the Spanish UPC Campus Manresa. For more related information, see the works of Tarrés, M. et al., 2021 [13,14]. 

In our previous paper [13], we described the idea and motivation behind the project, the methodology, the implementation plan, the participants in the project and their experience with STEAM projects, as well as preliminary results in relation to the preliminary design of the Qui-Bot robot. The Qui-Bot challenge oral presentation was related to [13] and addressed to Track 3. Bridging the diversity gap in STEM and is available at https://youtu.be/GQUIYO1sBxo (accessed on 30 April 2022). The preliminary design of the web interface monitoring the robot was added to our work in [14]. 

In this paper, we present the final design of the Qui-Bot robot, the implementation of the software interface (for monitoring and programming the robot), the running of the project, and the results.

### 1.3. Project Work Plan

The project work plan was previously published in references [13,14]. The Quí-Bot-H_2_O project is structured according the following phases:

Phase 1: Preparation for the activity with robots and sensors, including technical advice and equipment for the handling and controlling of the robots; the design of a set of activities with personalized robots by age with their corresponding chemical experiences and programming challenges adapted to the Spanish academic curriculum of chemistry and technology in (1) Early Childhood Education [15], (2) Primary Education [16], and (3) Compulsory Secondary Education [17]. In Section 2, we describe the hardware, software, and designed activities in detail.

Phase 2: The testing and fine-tuning of activities in controlled environments through expert advice using the Qui-Bot software, with the focus on attracting girls into science and technology. In Section 3, we describe the resulting test activities of the Qui-Bot H_2_O project in boys and girls in early childhood education and in k12 education.

Phase 3: Assessment of the results (classified in terms of age, gender, and social environment) and evaluation of the quantitative and qualitative impact of the activities before and after fulfillment. Mechanisms for evaluating quantitative and qualitative impacts will be designed and implemented appropriate to the characteristics of the project, format, and target audience. In order to evaluate the activities, data is collected from all the participants, the teachers involved, and the centers where these activities are carried out. For this, suitable survey models are designed. A set of variables are analyzed, both quantitatively and qualitatively. In the case of early childhood education students, data is extracted based on observation information relating to the boys and girls during the execution of activities with educational robotics applied to chemistry following the indications of the GRENEA group experts (see Section 3.1). In the case of primary education and high school students, the information that will be extracted is based on the two types of questionnaires detailed in Section 3.2).

Phase 4: The evaluation of the results from Phase 3 and the application of corrections in the activities and the design of robots according to the feedback provided from executing the project and according to Phase 3.

Phase 5: The dissemination of activities and the use of technological means for the dissemination of activities (i.e., a promotional video [18], the OCW website [19], brochure design, and dissemination material). The dissemination will be accompanied by talks by experts that give visibility to women in the STEM areas involved, for example, Nuria Infiesta, COO and Founder of Yasyt Robotics [20], an expert company in social robotics with numerous projects for the assistance and telepresence of dependent people with voice recognition devices.

Phase 6: The implementation of the project in classrooms through the training of teachers in schools and institutes. The training seeks to provide teachers and instructors with the tools that facilitate the use of the material, the application of the project, and even adaptation according to the educational project of their center. 

Phase 7: The evaluation of results and feedback by stakeholders after executing the project in classrooms. 

### 1.4. Teamwork

The Qui-Bot H_2_O team is composed of 15 entities and institutions following the quadruple helix model (open innovation 2.0) and 22 members (59.1% women). The preparation and testing of the activities were developed, from a technical point of view, by the Research Faculty of Chemical Engineering, Electronics, ICT, and Mathematics in the Department of Mining, Industrial, and ICT Engineering [21] at the Universitat Politècnica de Catalunya—BarcelonaTech (UPC) [22]. The pedagogical adaptation for the different age groups was carried out in collaboration with experts in the field attached to the GRENEA research group in education, experimentation, and learning [23] in the Faculty of Social Sciences at Manresa (UVic-UCC) [24].

## 2. Materials and Methods

The Qui-Bot project provides the hardware for mounting the robot (including all the steps to mount the Lego Mindstorms EV3 robot and the 3D design for the additional pieces), the software for controlling it, and the activities tailored to the Spanish curriculum. These teaching materials are all non-commercial and free for the educational centers who will participate in the project. All materials (hardware, software, and activities) can be downloaded from the OCW of the project [19]. The Qui-Bot robot allows for the performance of real chemical experiments with the goal of bringing chemical engineering and programming closer to students with daily examples. The project is designed to attract children and young people between preschool and primary education into computer science. Chemical experiments reproduce small-scale robots used in Industry 4.0, simulating basic industrial processes, without risk and danger to users. At the heart of the project are configurable and programmable robots and the necessary elements for a chemistry laboratory: robotic pipettes, liquid containers, and other items. The activities require common equipment (PCs, tablets, mobile phones) to control the robot. The scientific-chemical materials are easy to use and are without any risk or danger. These configurable and programmable robots complement the activities of the academic curricula of the different educational stages. The experiments are designed to provide a basic understanding of phenomena such as light reflection, the transmission of electricity, or the change of states and chemical reactions such as oxidation or fermentation through simple experiments. 

The Qui-Bot robot prototype presented here was inspired by the work of Stanford University’s Ingmar H. Riedel-Kruse [12]. The following improvements have been made to the work reported by [12]. As part of the Qui-Bot creation hardware, cuvette holders and additional LEGO pieces not included in Mindstorms EV3 were designed and printed in 3D. Thus, replicable products in educational centers will be more affordable. In addition, we use an infrared sensor to control the robot via a simple remote control. In our case, the robot connects with the PC, tablet or phone via a Bluetooth wireless connection in real-time. Lastly, we included a loudspeaker that allows the color sensor to verbalize the color result. Details of the robot’s hardware implementation are discussed in Section 2.1.

As for the novelties in [12], in the Qui-Bot software, a new web environment was designed to program and monitor the robot and attract both boys and girls to computational thinking. The Qui-Bot software does not used the rapid interface creation tool provided by Lego (Lego Mindstorms EV3 Home edition software) and used by [12]; it has been designed based on the specifications provided by experts in early childhood education, and to avoid reproducing gender stereotypes, and by emphasizing simple visual representations and easy-to-use features. It has been ensured that the software will be easy to install by teachers who will replicate the activities in schools. By using the Qui-Bot software, you can communicate with the robot and access it via a QR code on any Android phone or tablet. In Section 2.2, we describe the software implementation of the Qui-Bot interface. Additionally, the full implementation of the Qui-Bot software is provided, so it can be updated and adapted to the needs of the educational center that participates in the Qui-Bot project. Educational centers will be able to download the developed software free of charge through the OCW [19], and we welcome other robotics and science researchers to use, disseminate, and further develop the Qui-Bot robot project through open-source software and protocols.

Planned activities, unlike those conducted in [12], allow for the learning of computational language. The developed software interface allows for programs that consist of pressing a sequence of buttons that must be thought of a priori to be carried out. These buttons correspond to instructions that the robot executes in real time and the results of the instructions is very visual. Therefore, we can say that computational language is learned intuitively through trial and error. Furthermore, activities include small social challenges to increase the interest of girls in technology. The robotic scientific experimentation that is possible with Qui-Bot is described in Section 2.3.

### 2.1. Hardware

The Qui-Bot robot can be built from a single Lego Mindstorms EV3 kit and the following additional parts: the infrared sensor, RGB sensors, two large servo motors (pipette module and top module), and one medium servo motor (moves the robot cart), which contain internal rotation sensors. The design includes a 1 mL syringe, 20 standard 4.5 mL PMMA cuvettes, and a bucket holder designed using 3D printing. The sensor and infrared beacon allow for the remote control of the robot. The color sensor is used to reset the robot and position the colors in the buckets. In addition, a microSHDC card, USB connection, and hardware with Low Energy Bluetooth protocol are required. Figure 1, Figure 2 and Figure 3 show the main components. The RGB sensor allows for the discrimination of the color resulting from the mixture of colors and is used to relocate the robot to the first pipette to be analyzed. The infrared sensor allows for the robot to be governed, making it move to the right, to the left, or to raise and lower the syringe and the syringe plunger. 

### 2.2. Software

In this work, we present a novel implementation of a user interface in comparison to works by Ingmar H. Riedel-Kruse [12]. Using open-source tools, the Qui-Bot software is customized ad-hoc to control the robot. The interface of the robot was written in MicroPython EV3 [25], allowing for a more intuitive interaction with the sensors. Qui-Bot is cross-platform and is compatible with several platforms including Linux, Windows, and Mac OS. The web interface was built using Flask [26] and w3.css [27]. Moreover, it is compatible with the latest versions of the most popular browsers, such as Firefox, Edge, and Google Chrome. Figure 4 illustrates an example of how the Qui-Bot interface is implemented to allow access through the reading of a QR code or by reaching the robot’s IP address.

Figure 5 shows the interact tab, which permits you to control the robot by pushing buttons.

This panel is used to move the cart from left to right, so that the buckets can be positioned under the pipette. To transport liquids, you can use the pipette’s interface to go up, down, and perform a suction. With each of the beacon buttons, the infrared sensor is activated in command mode, and a pop-up window shows the instructions to the robot. Once the read button is pressed, a color sensor is activated, allowing the robot to read the current cuvette and to pronounce the color through its built-in speaker. The robot has a reset button that places the cuvette in the first bucket when it is pressed. The control panel on the program tab in Figure 6 allows you to play a series of instructions and focus on computational thinking: instruction sequences, loops, and computer logic. The built-in program is configured and displayed on the screen where the results of the execution of the program are reproduced. In this way, the robot is able to perform visual actions.

### 2.3. Pre-Programmed Chemical Experiments

The interface provides three pre-programmed chemical experiments that can be adapted to educational stages: (1) A dilution series, (see video available at: https://t.ly/5Bdo), (2) mixing, (see video available at: https://t.ly/uaym), and (3) density layers, (see video available at: https://t.ly/r2uU). The control panel in Figure 7 is used to select the pre-programmed experiments.

The following is a brief description of each pre-programmed chemical experiment. Detailed information on pre-programmed experiments is available in reference [28].

#### 2.3.1. Dilution Series

A serial dilution is the progressive reduction, bucket by bucket, of the concentration of a substance in solution, in this case dye. In general, the factor of dilution at each step is constant. One of its most important uses is in preparation of vaccines. When injected, a vaccine to a person must be diluted to the correct proportion to deliver an adequate viral load. Otherwise, if the vaccine is injected directly undiluted or with an incorrect dilution, adverse effects may develop.

To perform this experiment, six empty buckets need to be placed inside the black support block. The first bucket is filled with dye at 80% capacity and the next five with water up to 60% capacity. Then, the support is placed on the robot cart between the two blue pieces that act as guides. It is important to check that the pipette and the first bucket are aligned.

What the robot does is to move the solution from one bucket to another. Every time that this happens, the solution will be diluted with respect to the previous one until it is almost transparent. This experiment is run from the dilution series tab or can be programmed from the program tab using the control panel.

#### 2.3.2. Mixing of Primary Colors

A color is considered primary when it cannot be obtained with the mixture of any other color. The traditional primary colors are red, yellow, and blue. The primary color mix is used in everyday life very often.

Depending on the model, there are many applications. For example, the CMY model is used in printing, photography, and other activities. The RYB model, on the other hand, is used in art, drawings, and color palettes.

To perform this experiment, six empty buckets are placed inside the colored support block with white paper glued to the back. The first bucket is filled with blue and the third bucket with yellow at 80% capacity, with the rest of buckets remaining empty. The support on the robot cart is placed between the two blue pieces that act as guides. Finally, it must be checked that the pipette and the first bucket are lined up.

What the robot does is look for the two primary colors in the buckets. It then mixes them and obtains a new secondary color. This experiment is run from the primary color blending tab or can be programed via the program tab using the control panel.

#### 2.3.3. Density Layers

Density is a basic property of any liquid and is defined as the mass for a unit of volume. The densest liquid is the one that weighs the most and therefore the one that is deposited at the bottom of the bucket, with a liquid with a lower density being deposited on top.

This phenomenon can be observed in real life when bathing in a river or in the sea. Seawater, with a higher salt concentration, is denser than the water of a river. This is the reason that makes we float more easily when swimming in the sea than when swimming in a river. In the case of the Dead Sea, which has a salinity of 33.7%, it is possible to float without any effort.

To perform this experiment, six empty buckets are placed inside the colored support block. Then, 18 g of salt is diluted in 50 mL of water in a glass and added to the colored dye blue. Then, 6 g of salt is diluted in 50 mL of water in a glass and added to the colored dye red. The first cuvette is filled with the blue solution, the second cuvette with the red solution, and the third bucket with yellow water at a capacity of 90%, with the rest of buckets remaining empty. The support on the cart of the robot is placed between the two blue pieces that act as guides to ensure the first cuvette is aligned. What the robot does is deposit all the solutions in one bucket carefully. As the solutions have different densities, the solutions do not mix with each other and are distributed accordingly. This experiment is run from the density layers tab or can be programed from the program tab using the control panel.

### 2.4. Activities Design

The tested project activities tested are designed to enhance creativity and critical thinking skills, curiosity, resilience, problem-solving, resourcefulness, collaboration, and confidence. The following educational methods were followed in the design of the activities and materials:Learning through games (game learning): the scientific method and chemistry are learned from guidelines and through small challenge-games that teams can share in competitions;Project-based collaborative learning: students, without realizing it, become designers of robots that allow greater precision in chemical processes by witnessing the direct application of their robot in a chemical experiment, fostering creativity;Learning by doing: the fact that they can modify the experiment and the robot based on trial and error helps the children to develop the ability to think, experiment, and work;The improvement of the ability to solve problems: the design of the robot allows children to think like a professional and improve their problem-solving skills, thus encouraging creativity.

This program aims to encourage more girls into STEAM by relating activities to mini social challenges in which the children help the Qui-Bot robot achieve a particular goal, for example, helping Qui-Bot analyze water samples in order to detect viral plagues through colors in water solutions in order to curb possible pandemics. A sample activity script for a group of 30 girls, which is described in the next Section 3.2, is available at ttps://t.ly/tZgPh. 

## 3. Results

In what follows, we describe the resulting test activities of the Qui-Bot H_2_O project performed in two independent studies in small groups. In the case of early childhood education, we provide the results of activities fulfilled in the space for children named Lab 0_6: Discovery, Research, and Documentation Center for Science Education in Early Childhood [29], located within the Faculty of Social Sciences in Manresa (UVic-UCC). For girls from 7 years, we provide the evaluation of results achieved in the Manresa Technical Museum [30], with the help of experts in teaching educational robotics from CodeLearn. The pilot face-to-face activity has an introductory directed foundation (in which what the children test works for them and encourages them to achieve goals) and a complementary creative activity that allows them to adapt to various scenarios and encourages interaction between participants and teachers. All families gave their consent for the children to participate in the Qui-Bot H_2_O activities of the project.

### 3.1. Evaluation Results of Testing in Early Childhood Education

The first study was carried out for boys and girls between 3 and 6 years old. A total of six children participated in the study, two boys and four girls. The experience was developed in groups: a mixed group of one boy and one girl (of 4 and 5 years, respectively), two girls (both of 3 years), and one girl by themselves (of 5 years) and one boy by themselves (of 6 years), with all being guided by the educator and each observation lasting approximately 45 min. Among the possible activities, the exercise of mixing liquids to find out the resulting colors, and the reverse process, how to obtain a color from two others, were chosen. So, for example, the boys and girls had to plan in advance the sequence of buttons to push in order to obtain the green color in a new pipette from the colors blue and yellow. See Figure 8.

#### Analysis of Results of Testing in Early Childhood Education

The actions of the children were observed and registered/recorded with a video camera for later analysis. Oral questions were also asked at the end of the activity in order to assess the children’s level of satisfaction. The analysis of the results of the test was performed by experts of research in education and learning from the GRENEA group. The interactions between the computer and robot did not seem to cause a problem once the commands were understood (initially, some of the children had problems, but once they understood the commands, these were resolved). Some children had doubts about using the mouse and touch sensor. A girl, for example, tried to click directly onto the computer screen. Boys and girls seemed to show interest in the action by using their hands to grab pipettes, touch the screen, and touch the wheels of the machine. They showed interest in the proposal, as they were not focused (in general) on how long the activity lasted. In one case, one of the children was impatient and distracted towards the end. Some children found the planning of the sequence of activities of various movements (the creation of the first program) more difficult. The interaction between children in groups of two was considered appropriate in the proposal. During the activity, these pairs cooperated with each other correctly and there was intense communication and planning in terms of the organizing of roles between them. They were 100% sure that they liked it and that they would repeat it. They also expressed that they would like the robot presented to walk or move in some way. One of the things that struck them was that the robot named the resulting color. One of the girls verbalized that the chemical experiment robot was a machine, not a robot. Since the machine does not resemble a humanoid robot in any way, this is an important consideration to take into account. It does not appear to be a relevant differentiation in terms of interest in the activity between boys and girls.

As a result of the experience, the tool was improved so that the maximum permissible syringe cannot be moved further and that if the syringe is inside a pipette, it cannot be moved to the right or to the left. In general, the experience was defined as positive and the robot was not a problem. Making sequences was harder for the children. In order to make the tool more visual, it was proposed to design containers of greater capacity and to fix them. The interface was redesigned in a way that is responsible and so that all the icons are displayed on a single screen and a universal design for left-handed users was created. A new simplified visual redesign of images in the software was achieved to avoid stereotypical representations of scientists. Additionally, the redesign shows a gender equity and more diversified view of science, in conjunction with a more simplified visual representation without excess of visual inputs.

As a future work for new tests in early childhood education, it is important to consider how to integrate the interface so that it does not generate false expectations in children, for example, the creation of a more anthropomorphic robot. Further, it is necessary to ensure that in the testing there are no biases in the questions aimed at children. Additionally, it might be necessary to include as part of the learning the opportunity for participants to assemble other Lego constructs. Besides promoting practical tasks and rethinking applications, it would make more sense (1) for the children to use robots to do mixings and (2) to relate the mixes to more interesting practical outcomes. For example, the robots could be allowed to dye clothes, use kitchen dye and Chinese ink, or use chemicals that stain or bleach, so that it makes sense for children to use machines instead of their own bare hands to do the mixes, with the providing of examples of experimentation (e.g., fermentation type).

### 3.2. Evaluation Results of Testing in k12 Education

The second study was conducted to coincide with International Women’s Day in Science on 12 February. The activity was open to 18 girls between 8 and 13 years old. However, applications for the activity tripled in just one week, so it was decided to open the activity to a capacity of 30 girls. Of these, only 25% expressed having previous experience in one-off robotics activities. To carry out the activity, it was necessary to replicate the prototype design of six robots. Visual assembly steps for robot prototypes can be downloaded at OCW [19]. The activity was organized in terms of three groups of girls, each with two monitors, and there were 10 girls in each group. The results showed that this is a sufficient number to carry out the activity successfully. The groups of girls were sorted into participants between 8 and 9 years old, between 9 and 11 years old, and more than 12 years old. 

The experience was contextualized through a social challenge in which the girls had to help the robot, named Qui-Bot H_2_O, to solve a specific problem. The activity was supervised by six monitors (four women). In the experiment, the girls manually mixed basic colors with the robots, water, and food dyes (blue, yellow, red). For further information, please refer to the video [18].

More specifically, during the course of the activity, the attendees were able to: (1) perform a real-life chemical experiment based on color mixing and understand the results, (2) see how a machine (the Lego Mindstorms robot Qui-Bot H_2_O) is able to perform the same chemical experiment, (3) build, as a team, a replica of the Qui-Bot H_2_O robot using the numbered and detailed assembly instructions provided (see Figure 9), and (4) program the Qui-Bot H_2_O robot to reproduce the experiment (see Figure 10). The activity script is available in the OCW [19]. The activity lasted 2h and it was closed with the delivery of a specific diploma handed out by those responsible for equality policies in public administrations (City Council of Manresa [31], and county council of Bages [32], both in Manresa, Spain).

The validation was carried out by means of the direct observation of the monitors and in a questionnaire of 10 questions, with six of them being validated by means of the Likert scale according to interest in the activity (strongly disagree, little interest, enough interest, interest, strongly agree) and four of them being open questions.

#### 3.2.1. Analysis of Results of Testing in k12 Education

Two questionnaires were given to students at the end of the activity. We collected 27 responses to the questionnaires by hand. Three questionnaire results were lost. The answers to the questionnaires are available at https://t.ly/4uIq (accessed on 25 April 2022). The analysis of the results was performed with the help of the authors’ expertise in mathematics.

##### Analysis of k12 Users’ Achievement of the Activities

The aim of the first questionnaire was to rate the difficulty of the activity and the degree of satisfaction. Table 1 lists the six questions related to the project activities. 

The user’s sense of accomplishment associated with the activities and their level of motivation was collected via the Likert scale. Students marked their level of agreement or disagreement (1: strongly disagree, 5: strongly agree) with five predefined items. In what follows, the arithmetic means of the samples are denoted by $\bar{x}$, the standard deviations of the samples are noted by *s*, and the sizes of the samples are noticed by *n*. Table 2 shows, for each question, the frequency distributions in Likert scale items. 

Table 3 shows the distribution of the percentages of the Likert scale items. The results indicate a strong motivation for robotics after doing the activity. The confidence interval with 95% confidence is denoted by CI. The confidence interval for the question Q2 (overall rating of the course) on a scale 1–5 was (4.45; 4.89) with 95% of confidence. The results show a very high appreciation in terms of the degree of satisfaction with the activity carried out. The participants rated their intention to repeat the activity (Q6) with a CI (4.49; 4.94, which is higher than their initial expectations. They identified their previous interest (Q1) in the activity with a CI (4.20; 4.72), and they highlighted a very high-level interest in repeating the activities of the Qui-Bot project’s chemical experimentation with robotics, although they recognized that the activities were not trivial. The ease of the activity (Q4) was rated with a CI (3.08; 3.58). And, finally, the users would have liked to be able to dedicate more time to the activity (Q3), with a CI of (3.92; 4.74). When asked specifically which of them had done robotics before (Q5), less than 20% affirmed having previous experience with robotics, with a CI (1.96; 3.07), despite the wide availability of activities in this sector. Thus, the interest aroused in this activity was significant, even for children not used to performing tasks with robots. 

On Table 4, we can see the hypothesis tests of the means, shown by µ, for questions Q1 to Q6, based on the girls’ age, whether they were younger than 10 years old or older than 10 years old. This type of test is performed using a t-Student test, previously observing whether the variances, symbolized by σ^2^, can be considered equal or different. Therefore, it is necessary to perform a test of two variances before the test of means. Obviously, the necessary information for this entire process is given by means of the samples, denoted by $\bar{x}$, the standard deviations denoted by *s*, and the sample sizes *n*. The *p*-value is a number that we have to test against our significance level α = 0.05. Thus, when α ≤ *p*, we accept the null hypothesis, that is, equality of variances or equality of means. We see that, in all cases, the variances and the means can be considered equal, working at a level of significance α = 0.05, that is, a 95% degree of confidence. Regarding the results of the hypothesis test shown in Table 4, we can conclude that the questions classified by age do not present significant variations because their *p*-value (*p*) is greater than 0.05 (level of significance). Therefore, there is no significant difference between girls up to 10 years old and those under 10 when it comes to their responses to the six questions. In the following, we compute the response rate to each individual question and according to each Likert scale item.

Table 5 and Table 6 show the response proportions for each type of question in percentages by age classified according to Likert. In Table 5, the results for girls younger than 10 years old are presented, and in Table 6 shows those for girls older than 10. 

The hypothesis test of contrasted proportions, displayed in Table 7, evaluated the two groups of ages considered above (greater or equal to 10 years old, less than 10 years old). For a *p ≤* 0.05, it is considered that there are significant differences in responses between the two age groups. The following represents the explanations of the three cases reflected in Table 7. According to the test, the proportion of girls under ten years who answered “little interest” in Q5 is less than the percentage of girls of the same age or older. According to the answer “interest” to the question Q1, the proportion of girls under 10 years is lower than the proportion of girls over 10 years of age. In Q1, we can report that the proportion of girls under 10 who selected “strongly agree” is higher than that for girls older than or equal to 10 years of age. A *p*-value of greater than 0.05 indicates no statistically significant difference in age groups between the questions and/or levels of Likert scales in the other cases.

Lastly, we measured the degree of dispersion for the answers to the key questions.

Figure 11 illustrates the underlying distribution of responses according to the Likert scale (1: strongly disagree, 5: strongly agree): Q2 (I liked the activities) and Q6 (I want to repeat them). In response to question Q2, most people enjoyed the activity. With regard to question Q6, most of the answers collected indicate that most people were interested in repeating the activity. With regard to questions Q2 and Q6, the responses are dispersed differently. The answers to question Q2 are more dispersed than those to question Q6. For question Q2, the answers are more dispersed (between values 4 and 5) than for question Q6 (between values 4.5 and 5).

Figure 12 compares the perceptions of the ease of the Qui-Bot activities (Q4) with the results relating to users not having done robotics activities before (Q5). An overwhelming majority of respondents answered between 1 and 3 when asked if they had previously worked with robotics (Q5). A majority of responses (Q4) were between 3 and 4 when it comes to feeling at ease during the activity. A low number of girls had participated in robotics activities, despite the wide availability of robotics courses. This fact highlights and reinforces the idea that technology and robotics need to be made more accessible to these ages and attractive to girls.

##### Gathering Information from k12 Students

In the second questionnaire, the participants were requested to answer the questions in Table 8. When asked what part of the activity they liked the most, the results exhibit the girls’ interest in (1) the robot assembly hardware, (2) the robot programming software and how the robot carried out the experiments, and (3) chemical experimentation by hand. The results are as follows: 41% (11/27) took an interest in mounting the robot, 41% (11/27) pointed to robotics programming and how the robot made the experiments, and only 18% (5/27) preferred color-mixing manually. Regarding the question about why they signed up for this activity, 37% (10/27) suggested because a friend had signed up or because they found it fun that only girls were allowed to sign up, and 63% (17/27) gave vocational motivational reasons such as “I want to be a scientist”, “I want to do experiments”, “I like to assemble things”, “I am interested in robotics”. If we reference preferred school subjects, only 18.5% chose mathematics, biology, or technology, with art and physical education being the star subjects in 81.5% of the cases. Considering the previous results relating to activity satisfaction levels, it can be inferred that the inclusion of the Qui-Bot activities in science and technology subjects would mean an increase in satisfaction in STEM subjects.

Finally, 44% of the girls responded that their favorite game was linked to video games, 18.5% marked all the options: sports games, board games, and video games, and 29.6% selected sports games. Therefore, 62.5% preferred games linked to computer programming. The girls came from 11 different primary schools (three concerted schools) and four outside the city, where the activity took place. No significant differences were observed according to the school where they come from.

## 4. Discussion

As a result of the tests, the next steps are to design and improve a new Lego-based mobile robot prototype that performs chemical experiments and includes other sensing elements that allow voice recognition through simple and easy-to-use commands to facilitate robot–child interactions. On the other hand, it is proposed that the connection between the robot and the computer is improved through the use of Wi-Fi, which would allow faster interactions and the monitoring of the robot, and possibly also concurrent access. The design is to be extended so that it can be controlled through a mobile application. Further research is being done on the management of a remote laboratory, so that, for example, the robot located at the school lab can be commanded from home. Finally, work is being done on another type of non-Lego-based robot, a more humanoid prototype, that will allow the project to be implemented at a lower cost, and its validation against the proposed solutions is expected. Finally, we are working on the design of an application that will help monitor the results of the surveys and which can have a greater educational impact. A new time validation activity of more than 2 h is being prepared in the form of a hackathon/contest competition to take place throughout Catalonia, in which boys and girls will participate, and that will reward the completion of the challenges launched using the Qui-Bot H_2_O design. Future work includes the development of an ethical code for the use of robots in the classroom and a reflection on how the relationship between children and robots enhances their creativity.

## 5. Conclusions

Qui-Bot was developed to bridge the gender gap in ICT-related fields and to address STEM employment requirements by integrating women into IT and tech-related fields. Compared to Chemistry, the University of Politècnica de Catalunya—BarcelonaTech [33] shows that fewer women are enrolling in ICT engineering systems and computer engineering. Table 9 compares women enrollment data collected from two campuses within the same university between 2017–2022 in relation to computer systems engineering and chemical engineering. In order to introduce girls to computational thinking and robotics, Qui-Bot H_2_O takes advantage of their interest in engineering, especially in the chemical industry.

The goal of our work is not to determine whether boys and girls interact differently with Qui-Bot. According to the evaluation expressed, the activity was validated only with girls who exhibited a positive attitude and who remained attentive throughout the activity. Activities involving robotics tend to be more popular with boys than with girls, who rarely attend robotics workshops. Our new approach makes use of chemistry processes and social applications to reach out to girls’ interest in robotics. In this work, we demonstrate that robotics activities developed using these ingredients are attractive to girls. It would be interesting to analyze if gender influences behavior when facing the same activity in future work.

The downloadable software and web framework to control the robot is available at https://t.ly/jELD. Depending on the educational stage, the ad hoc software allows students to: (1) Replicate pre-programmed chemical experiments; (2) program the robot intuitively using programming instructions that the robot understands; or (3) monitor the robot through small steps mirroring computational thinking. The instructions to build the robot and the 3D design of non-lego pieces are available at https://t.ly/com6. The activities are based on (1) a dilution series, https://t.ly/5Bdo, (2) mixing, https://t.ly/uaym, and (3) density layers, https://t.ly/r2uU. The activities can be adapted for early ages and primary and secondary education and is freely accessible through the OCW [19] for use by other instructors. All of these activities are prepared to be fully adaptable to meet the specific needs of scientific concepts at all educational levels. We hope that the findings of this study will assist school lecturers in enhancing Qui-Bot activities. 

## Figures and Tables

**Figure 1 sensors-22-03719-f001:**
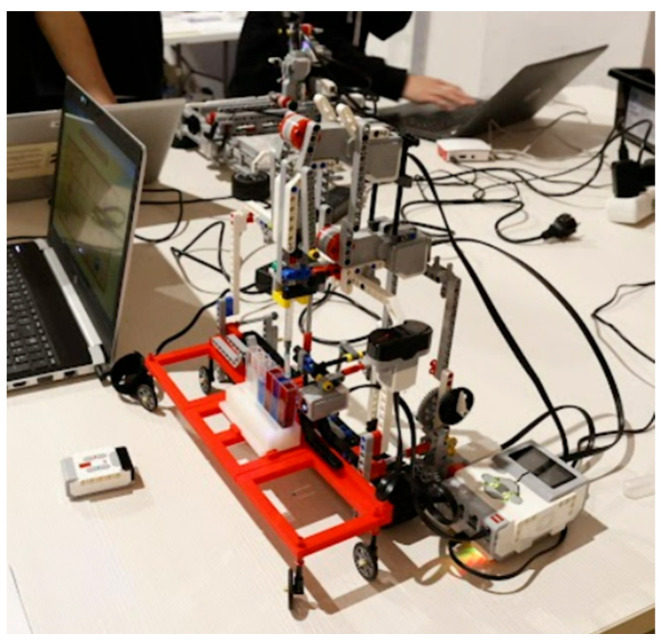
The Qui-Bot Hardware.

**Figure 2 sensors-22-03719-f002:**
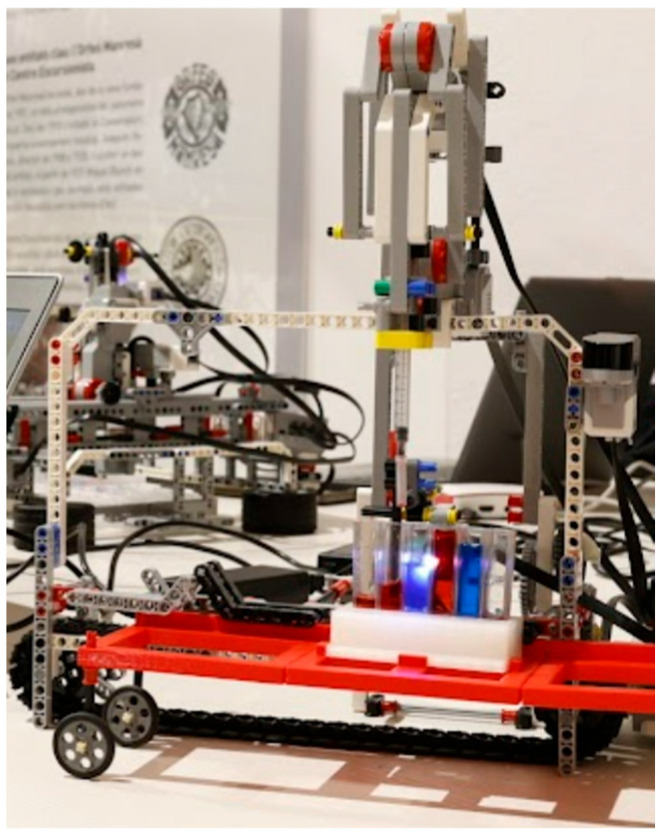
Qui-Bot with the RGB sensor activated (front view).

**Figure 3 sensors-22-03719-f003:**
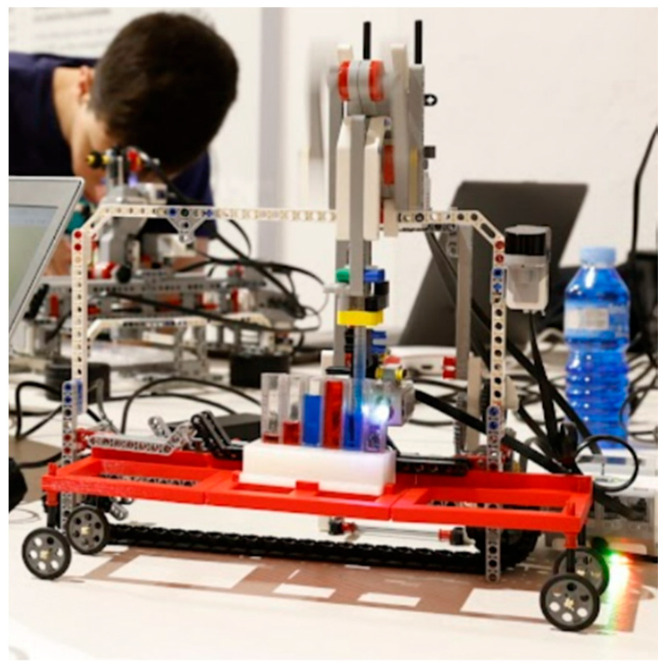
Qui-Bot with the syringe plunger, which it pulls up to suck the colored liquid in the pipette holder.

**Figure 4 sensors-22-03719-f004:**
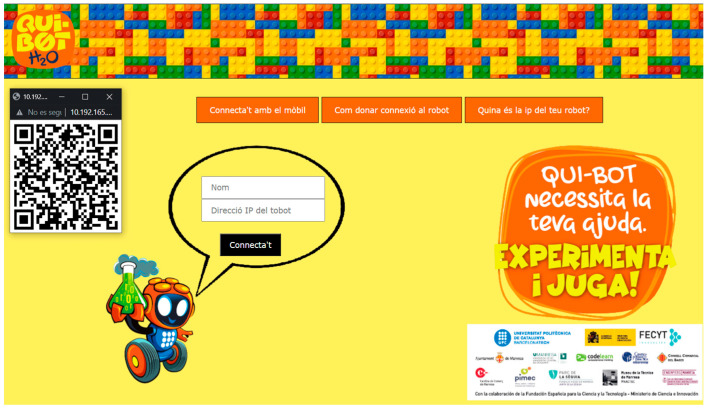
The web interface for Qui-Bot software.

**Figure 5 sensors-22-03719-f005:**
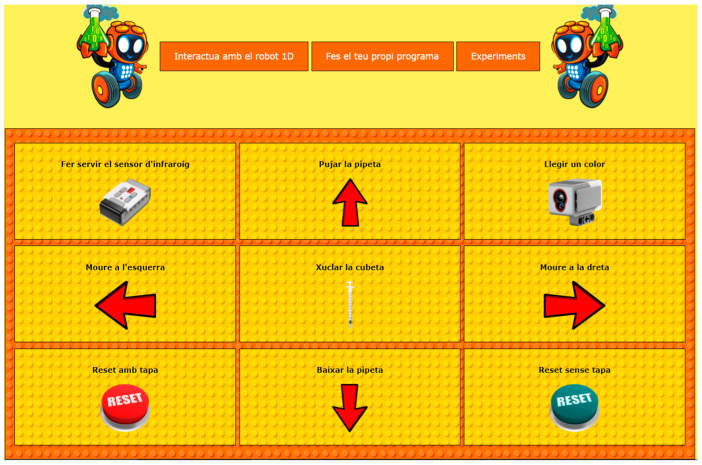
Interaction with the robot via web interface.

**Figure 6 sensors-22-03719-f006:**
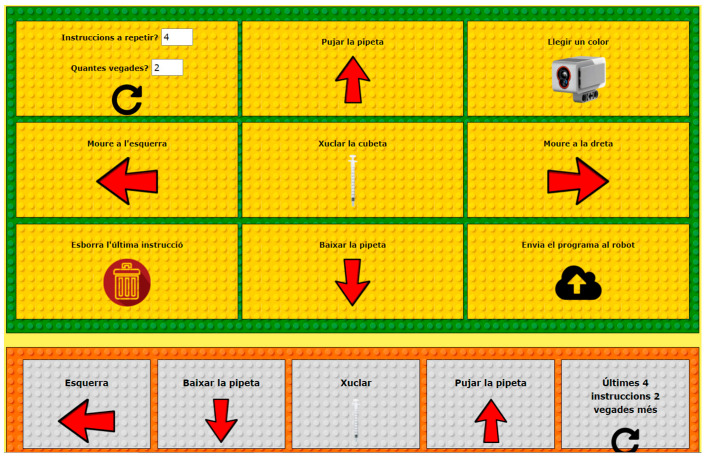
Program tab in the Control Panel that allows a user to program the robot.

**Figure 7 sensors-22-03719-f007:**
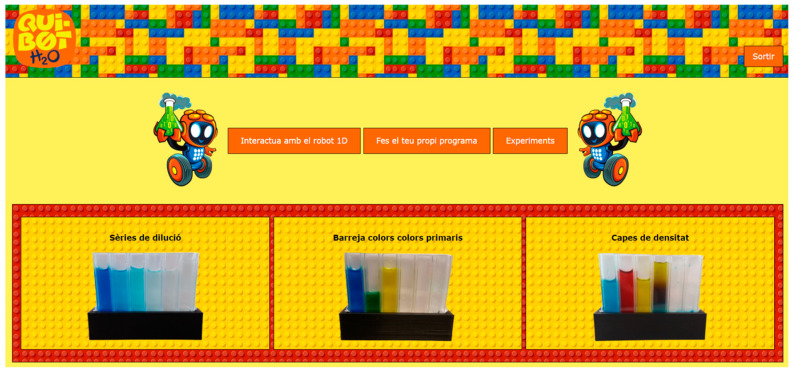
Control Panel to select pre-programmed experiments.

**Figure 8 sensors-22-03719-f008:**
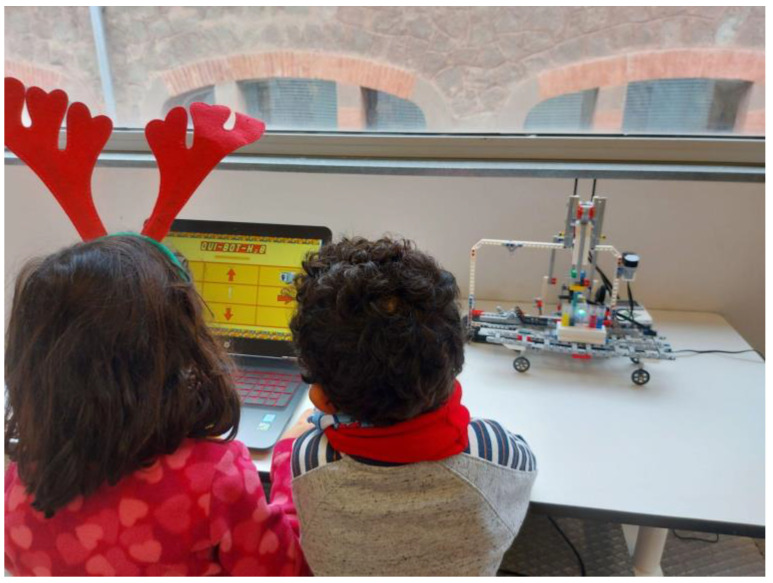
Testing Qui-Bot in early ages.

**Figure 9 sensors-22-03719-f009:**
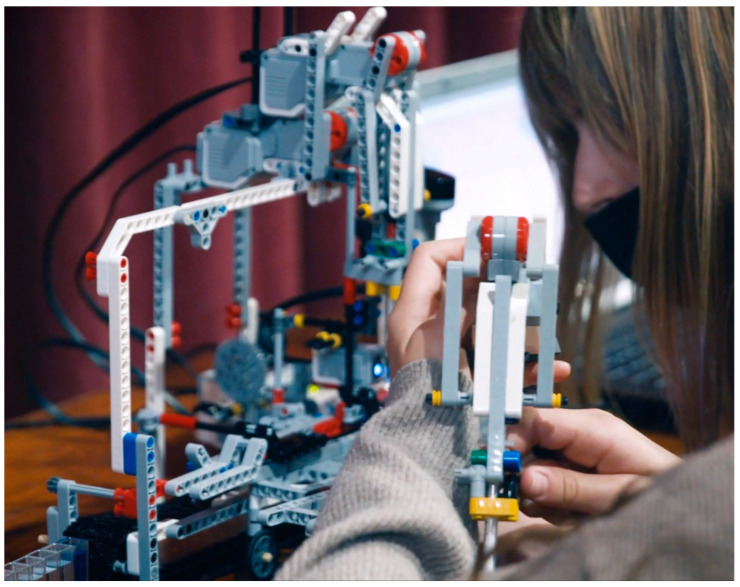
Testing Qui-Bot in k12: building the Qui-Bot.

**Figure 10 sensors-22-03719-f010:**
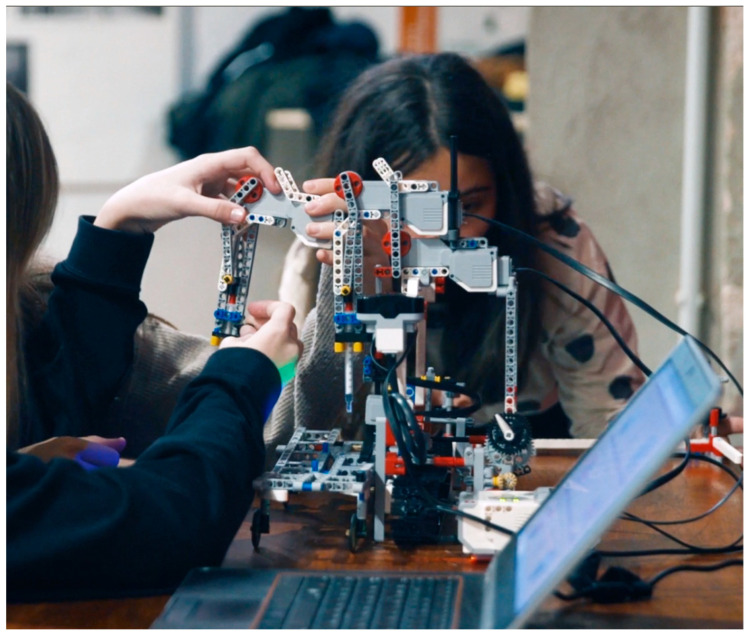
Testing Qui-Bot in k12: building and programming.

**Figure 11 sensors-22-03719-f011:**
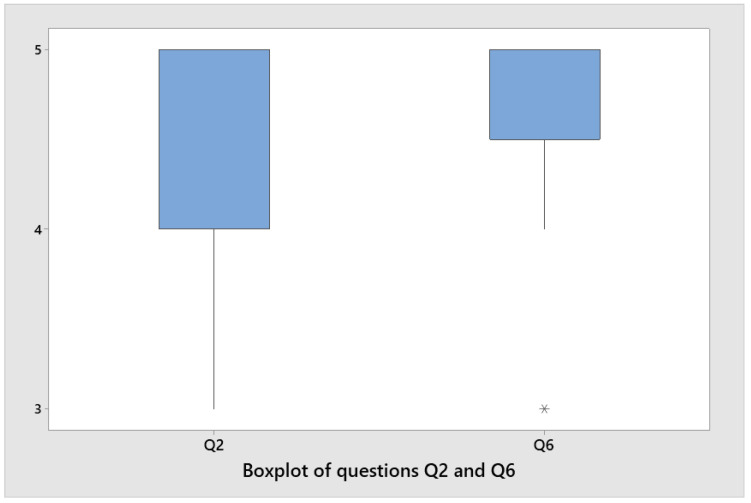
The boxplot for questions Q2 and Q6. In the figure shown in question Q6, value 3 has been marked as an outlier with an *, meaning that it corresponds to an atypical value (see value 3 in Table 2).

**Figure 12 sensors-22-03719-f012:**
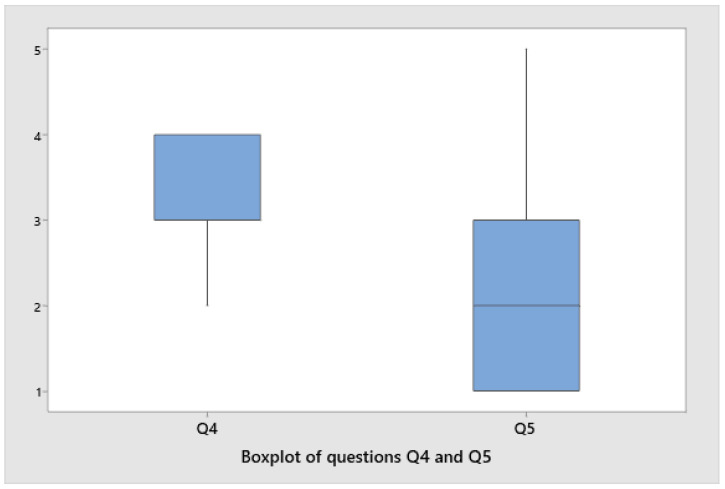
The boxplot for questions Q4 and Q5.

**Table 1 sensors-22-03719-t001:** Specific questionnaire.

Q1: I was impatient to do the activities of the Qui-Bot H_2_O project
Q2: I enjoyed the activities
Q3: The time to do the activities is enough
Q4: I find the activities easy
Q5: I have tried robotics before
Q6: I would like to repeat the activities of the Qui-Bot H_2_O project

**Table 2 sensors-22-03719-t002:** Frequency distribution table in Likert scale items.

		Question					
Items		Q1	Q2	Q3	Q4	Q5	Q6
1	strongly disagree	0	0	0	0	7	0
2	little interest	0	0	2	2	9	0
3	enough interest	2	1	5	14	6	1
4	interest	10	7	2	11	0	5
5	strongly agree	14	19	18	0	5	19
*n*		26	27	27	27	27	25
$\bar{x}$		4.46	4.67	4.33	3.33	2.52	4.72
*s*		0.63	0.54	1.02	0.61	1.37	0.53

**Table 3 sensors-22-03719-t003:** Distribution of percentages of the Likert scale items.

		Question					
Items		Q1	Q2	Q3	Q4	Q5	Q6
1	strongly disagree	0.0%	0.0%	0.0%	0.0%	25.9%	0.0%
2	little interest	0.0%	0.0%	7.4%	7.4%	33.3%	0.0%
3	enough interest	7.7%	3.7%	18.5%	51.9%	22.2%	4.0%
4	interest	38.5%	25.9%	7.4%	40.7%	0.0%	20.0%
5	strongly agree	53.8%	70.4%	66.7%	0.0%	18.5%	76.0%

**Table 4 sensors-22-03719-t004:** Hypothesis test of variances and means distribution.

Age	Q1	Q2	Q3	Q4	Q5	Q6
<10	$\bar{x}$ = 4.64 *s* = 0.72 *n* = 14	$\bar{x}$ = 4.73 *s* = 0.57 *n* = 15	$\bar{x}$ = 4.27 *s* = 1.12 *n* = 15	$\bar{x}$ = 3.40 *s* = 0.61 *n* = 15	$\bar{x}$ = 2.33 *s* = 1.35 *n* = 15	$\bar{x}$ = 4.77 *s* = 0.57 *n* = 13
≥10	$\bar{x}$ = 4.25 *s* = 0.43 *n* = 12	$\bar{x}$ = 4.58 *s* = 0.49 *n* = 12	$\bar{x}$ = 4.42 *s* = 0.86 *n* = 12	$\bar{x}$ = 3.25 *s* = 0.60 *n* = 12	$\bar{x}$ = 2.75 *s* = 1.36 *n* = 12	$\bar{x}$ = 4.67 *s* = 0.47 *n* = 12
	*p* = 0.105 $\sigma_{1}^{2}=\sigma_{2}^{2}$	*p* = 0.642 $\sigma_{1}^{2}=\sigma_{2}^{2}$	*p* = 0.398 $\sigma_{1}^{2}=\sigma_{2}^{2}$	*p* = 0.975 $\sigma_{1}^{2}=\sigma_{2}^{2}$	*p* = 0.933 $\sigma_{1}^{2}=\sigma_{2}^{2}$	*p* = 0.522 $\sigma_{1}^{2}=\sigma_{2}^{2}$
	*p* = 0.127 $\mu_{1}=\mu_{2}$	*p* = 0.496 $\mu_{1}=\mu_{2}$	*p* = 0.717 $\mu_{1}=\mu_{2}$	*p* = 0.543 $\mu_{1}=\mu_{2}$	*p* = 0.448 $\mu_{1}=\mu_{2}$	*p* = 0.654 $\mu_{1}=\mu_{2}$

**Table 5 sensors-22-03719-t005:** Response proportion of girls under 10 years of age.

		Question					
Items		Q1	Q2	Q3	Q4	Q5	Q6
1	strongly disagree	0.0%	0.0%	0.0%	0.0%	40.0%	0.0%
2	little interest	0.0%	0.0%	13.3%	6.7%	13.3%	0.0%
3	enough interest	14.3%	6.7%	13.3%	46.7%	33.3%	7.7%
4	interest	7.1%	13.3%	6.7%	46.7%	0.0%	7.7%
5	strongly agree	78.6%	80.0%	66.7%	0.0%	13.3%	84.6%

**Table 6 sensors-22-03719-t006:** Response proportion of girls equal to or older than 10 years of age.

		Question					
Items		Q1	Q2	Q3	Q4	Q5	Q6
1	strongly disagree	0.0%	0.0%	0.0%	0.0%	8.3%	0.0%
2	little interest	0.0%	0.0%	0.0%	8.3%	58.3%	0.0%
3	enough interest	0.0%	0.0%	25.0%	58.3%	8.3%	0.0%
4	interest	75.0%	41.7%	8.3%	33.3%	0.0%	33.3%
5	strongly agree	25.0%	58.3%	66.7%	0.0%	25.0%	66.7%

**Table 7 sensors-22-03719-t007:** Proportions hypothesis test results.

	Q1	Q2	Q3	Q4	Q5	Q6
strongly disagree	*	*	*	*	*p* = 0.062	*
little interest	*	*	*p* = 0.189	*p* = 0.869	*p* = 0.014	*
enough interest	*p* = 0.173	*p* = 0.362	*p* = 0.438	*p* = 0.547	*p* = 0.121	*p* = 0.327
interest	*p* 0.000	*p* = 0.095	*p* = 0.869	*p* = 0.484	*	*p* = 0.109
strongly agree	*p* = 0.006	*p* = 0.221	*p* = 1.000	*	*p* = 0.438	*p* = 0.294

* means that the compared proportions correspond to 0% values (Table 5 and Table 6), and Minitab does not provide a *p*-value in these cases. However, since the percentages are equal to zero, we have to assume their equality.

**Table 8 sensors-22-03719-t008:** Generic questionnaire.

Q1: Which is your favorite game (sport games, board games, video games)?
Q2: What subject is your favorite in school?
Q3: Why have you signed up for this activity?
Q4: What activity did you like the most today?

**Table 9 sensors-22-03719-t009:** A comparison of the percentage of women enrolling in engineering studies.

**Bachelor’s Degree in ICT Systems Engineering (UPC-EPSEM)**						**Bachelor’s Degree in Chemical Engineering (UPC-EPSEM)**					
	**2017–2018**	**2018–2019**	**2019–2020**	**2020–2021**	**2021–2022**		**2017–2018**	**2018–2019**	**2019–2020**	**2020–2021**	**2021–2022**
Women	6	2	3	6	2	Women	27	26	35	33	24
Total	51	43	39	54	40	Total	70	69	82	75	53
%Women	11.8%	4.7%	7.7%	11.1%	5.0%	%Women	38.6%	37.7%	42.7%	44.0%	45.3%
**Bachelor’s Degree in Informatics Engineering (UPC-FIB)**						**Bachelor’s Degree in Chemical Engineering (UPC-ETSEIAAT)**					
	**2017–2018**	**2018–2019**	**2019–2020**	**2020–2021**	**2021–2022**		**2017–2018**	**2018–2019**	**2019–2020**	**2020–2021**	**2021–2022**
Women	45	39	55	59	55	Women	49	44	41	38	33
Total	411	416	410	407	409	Total	134	132	122	112	83
%Women	10.9%	9.4%	13.4%	14.5%	13.4%	%Women	36.6%	33.3%	33.6%	33.9%	39.8%

## Data Availability

The answers to the questionnaires are available at https://t.ly/4uIq (accessed on 25 April 2022).

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
