# Peer review of "Sparking the Interest of Girls in Computer Science via Chemical Experimentation and Robotics: The Qui-Bot H2O Case Study †"

_sensors, 2022, doi:10.3390/s22103719_

Round 1
Reviewer 1 Report
The paper is an extended version of previous publications on the Qui-Bot project, which aims at motivating young (female) learners to engage with chemistry experiments through a robotic kit. The idea of the robot kit is based on another paper [12] and it seems the technical contribution of the paper is that the software system for conducting chemistry experiments are newly designed. A further contribution is that the robot and the robot control system has been tested in different age groups with up to 30 learners. The learners were to fill out questionnaires which are thoroughly analysed in the paper.
While I definitely support the idea to spark interest in young (esp female) learners for STEM, I have some problems with the paper.
1) the paper is an extended version, and I was not able to read the original papers to see the novel contribution
2) the differences to paper [12] needs to be pointed out in greater detail
3) While the paper describes an interesting experiment for motivating learners for STEM, I am wondering if this type of contents is appropriate for the Sensors journal, as it does not really fit the scope of the journal.
4) The project website is not working
These issues need to be addressed in a future revision of the paper.
Besides this, there is some editing required:
l 84 3,1 -> 3.1; 10 and 15 % -> 10% and 15 %
l 319: what does Lab 0_6: mean? is it 0-6?
p 11: OCW (See [30]) -> OCW
l 392: Results showed that was a sufficient number to be able to carry out the activity successfully. -> Results showed that it/this was a sufficient number to be able to carry out the activity successfully.
ll 401-407, the sentences in enumeration should be lower case.
l408: closured-> closed
l433: Frequency -> frequency
l459: why is the unequality typeset in red?
There are spurious additional white spaces throughout the document.
Author Response
First of all, thank you for the opportunity to revise our manuscript. We really appreciate our comments, in terms of opportune technical corrections and as clarifying observations. We have addressed our comments and we think that the paper has improved, and it is ready for publication.
Please see the attached file where we provide a point-by-point response to our comments. Please, do not hesitate to contact to us for additional comments or suggestions.

Reviewer 2 Report
Thank you for the opportunity to review this work. I find it both interesting and relevant. It describes a larger program where several institutions are involved and require strong group work.
There are some issues, however, that need to be addressed for the article to be publishable. I will try to be as clear and succinct as possible:
- There are some aspects of the study that are mentioned but not particularly explained. In the abstract, the authors mention this to be a multidisciplinary and interdisciplinary study. I suppose they refer to the fact that teaching Chemistry is aided by ICT (robots in particular). Maybe it also refers to the many disciplines of the members of the team. Maybe is both. Maybe it is something else. It is never clearly explained in the text.
- The objective of the work is to "develop an educational project" to incorporate and improve computational thinking. Moreover, a strong emphasis is made on the application of this model to girls to increase their interest in STEM. The gender gap and the need for STEM graduates are clearly stated in the literature review. However, it is never clear how the activities intend to address so, especially the impact on girls. What do you mean by "implement the computational thinking in the different educational stages."? how did you achieve it?
- The connection that is made to the sustainability goals is very shallow. It seems that is only placed there to try to provide greater relevance to the study. I am not saying it is wrong to mention them, but I believe further elaboration would be in order.
- Be careful with acronyms. You use IT for Telecommunications and Information Technology, and then ICT for information and communication technology. Be consistent. The latter is better but communications should be expressed in plural.
- In the second paragraph in section 1.2 it seems like 10% of all women in the world work in AI. Rephrase: "...10 to 15% of workers in AI are women".
- Section 1.3 is very confusing. It describes activities for project implementation that are disconnected and lack a sequence. Dissemination of information is before evaluating results, which in turn is before implementing the activities. Give those points coherence.
- How are these activities "analytic process and empirical experimentation". Merge this section with the method below, and integrate it to strengthen the description.
- Based on what premises were the activities designed? what is derived from the previous step?
- Who provided the feedback to correct the activities? Based on what?
- In general the method is described very lightly. More detail is necessary for this section to make it robust.
- If the software and the hardware are available commercially, I would assume that the contribution of the paper is the design of the activities, and the method to implement them. I believe that this is not strongly explained or detailed in the paper and needs to be improved.
- How were the results recorded, coded, and interpreted?
- The results are presented strangely. Differences in means for several groups should be addressed with ANOVA, while between two groups with t-tests. Nonetheless, the p-values are not shown (I am sorry if I failed to see them, in case they are there). I think box plots would depict very clearly if there are differences in behavior for these groups. The way the results are presented, it is dry and easy to get confused. Provide better ways to show your results.
- The paper is full of bad uses of the English language. I would suggest proof editing through a professional.
- The conclusions show bold statements about the effect of the experiment on girls, where there is no evidence that it should be different from that of boys. I suggest caution when making statements.
Overall, this paper presents the results of a gigantic effort, but it fails to do so in an effective way. It is paramount that the authors present clearly what is the contribution beyond those of the previous publications. I urge the authors to do an exhausting review of their work to make it publishable and give the study the justice that it deserves.
Author Response
First of all, thank you for the opportunity to revise our manuscript. We really appreciate your comments, in terms of opportune technical corrections and as clarifying observations. We have addressed your comments and we think that the paper has improved, and it is ready for publication.
Please see the attached file that contains the revision to the major points that were pointed out. Please do not hesitate to contact to us if more information is needd

Round 2
Reviewer 1 Report
The authors addressed all the issues appropriately.
Author Response
We are grateful for your consideration of this manuscript.
Reviewer 2 Report
Thank you for considering my suggestions. I believe there are significant improvements. I would still consider a couple of things:
- Including the word "girls" 13 times in the paragraph, does not indicate why the results and relevance of the study should be different from those of boys. Without wanting to pester you, I believe that you still need to make your case. Not because you only did your study with girls, it would demonstrate that things are different than with boys. I understand you don't have data to show it, but use your existing body of knowledge to justify the assumption.
- The boxplots are better than the tables to show differences, and are a lot more clear. The p-values in your tables actually indicate that most of your hypotheses don't hold. You don't need experimental rigor to show your point, but I will leave that up to you.
Thank you and good luck!
Author Response
Please see the attachment. We have addressed all the concerns raised. We thank the reviewer for their careful reading of the manuscript and their constructive remarks.
